# Intraoperative and Postoperative Effects of Dexmedetomidine and Tramadol Added as an Adjuvant to Bupivacaine in Transversus Abdominis Plane Block

**DOI:** 10.3390/jcm12227001

**Published:** 2023-11-09

**Authors:** Zeki Korkutata, Arzu Esen Tekeli, Nurettin Kurt

**Affiliations:** 1Department of Anesthesiology and Reanimation, Bingol State Hospital, Bingol 12000, Turkey; zeki.korkutata@hotmail.com; 2Department of Anesthesiology and Reanimation, Faculty of Medicine, Van Yuzuncu Yil University, Van 65080, Turkey; dr.nurettinkurt@gmail.com

**Keywords:** bupivacaine, transversus abdominis plane block, dexmedetomidine, tramadol, cholecystectomy

## Abstract

**Background:** We aimed to evaluate the intraoperative hemodynamics, opioid consumption, muscle relaxant use, postoperative analgesic effects, and possible adverse effects (such as nausea and vomiting) of dexmedetomidine and tramadol added as adjuvants to bupivacaine in the transversus abdominis plane block (TAP block) to provide postoperative analgesia. **Materials and Methods:** This was a prospective, randomized, controlled trial on patients who underwent laparoscopic cholecystectomy. After obtaining ethical approval at the Van Yuzuncu Yil University and written informed consent, this investigation was registered with ClinicalTrials.gov (NCT05905757). The study was conducted with 67 patients with ASA I–II physical status, aged 20–60 years, of either sex who were scheduled for an elective laparoscopic cholecystectomy under general anesthesia. Exclusion criteria were the patient’s refusal, ASA III and above, a history of allergy to the study drugs, patients with severe systemic diseases, pregnancy, psychiatric illness, seizure disorder, and those who had taken any form of analgesics in the last 24 h. The patients were equally randomized into one of two groups: Group T (TAP Block group) and Group D (Dexmedetomidin group). Standard general anesthesia was administered. After intubation, Group T (Bupivacaine + adjuvant tramadol) = solutions containing 0.250% bupivacaine 15 mL + adjuvant 1.5 mg/kg (100 mg maximum) tramadol 25 mL and Group D (Bupivacaine + adjuvant dexmedetomidine) = solutions containing 0.250% bupivacaine 15 mL + 0.5 mcg/kg and (50 mcg maximum) dexmedetomidine 25 mL; in total, 40 mL and 20 mL was applied to groups T and D, respectively. A bilateral subcostal TAP block was performed by the same anesthesiologist. Intraoperative vital signs, an additional dose of opioid and muscle relaxant requirements, complications, postoperative side effects (nausea, vomiting), postoperative analgesic requirement, mobilization times, and the zero-hour mark (patients with modified Aldrete scores of 9 and above were recorded as 0 h), the third-hour, and sixth-hour visual analog scale (VAS) scores were recorded. The main outcome measurements were the effect on pain scores and analgesic consumption within the first 6 h postoperatively, postoperative nausea and vomiting (PONV), and time to ambulation. The secondary aim was to evaluate intraoperative effects (on hemodynamics and opioid and muscle relaxant consumption). **Results:** It was observed that dexmedetomidine and tramadol did not have superiority over each other in terms of postoperative analgesia time, analgesic consumption, side effect profile, and mobilization times (*p* > 0.05). However, more stable hemodynamics were observed with dexmedetomidine as an adjuvant. **Conclusions:** We think that the use of adjuvant dexmedetomidine in the preoperative TAP block procedure will provide more stable intraoperative hemodynamic results compared with the use of tramadol. We believe that our study will be a guide for new studies conducted with different doses and larger numbers of participants.

## 1. Introduction

Today, cholecystectomy can be performed with open and laparoscopic techniques [1]. After laparoscopic surgery, most patients have severe abdominal pain and feel the need for effective analgesia [2]. Pain is a factor that causes a delay in postoperative discharge [3]. One of the most important goals of anesthesia applications is perioperative pain control. The development of different methods for pain control reduces the incidence of side effects due to high-dose analgesics. The use of locoregional anesthesia has spread during the last few years as it can be useful to reduce the time of hospitalization, the incidence of adverse events related to general anesthesia, and pain; moreover, it can be useful to achieve, almost, opioid-free anesthesia. It can be used as a substitute for general anesthesia or as an adjuvant to general anesthesia to achieve better pain control and lessen the use of intra- and postoperative analgesics (most all opioids), thus resulting in less nausea and vomiting risk, faster mobilization and recovery, and more rapid hospital discharge [4,5]. The use of ultrasonography (USG) increases the success of nerve blocks and plane blocks and reduces the amount of drugs used, as well as the possibility of side effects and complications [6,7]. The transversus abdominis plane block is one of the most frequently used regional approaches to provide analgesia in the anterior abdominal wall after laparoscopic abdominal surgery. Local anesthetic is injected into the planar space between the internal oblique and transversus abdominus muscles, and the afferent nerves are blocked [8]. By providing effective analgesia, this method significantly reduces the postoperative stress response and facilitates postoperative recovery [9].

Dexmedetomidine, a highly selective α2-adrenergic agonist that has sedative, analgesic, sympatholytic, and anesthetic effects, is an interesting agent that prolongs the block time with its adjuvant use with local anesthetics. Its central mediated analgesia, the mechanism by which dexmedetomidine enhances the quality of regional anesthesia when used as an adjuvant to local anesthetic, can be explained by two peripheral mechanisms. First is the vasoconstrictor effect around the site of injection, which leads to a delay in the absorption of the local anesthetic and prolongs the duration of the local anesthetic effect. The second mechanism is the direct action of dexmedetomidine on the activity of peripheral nerves [10]. Parenterally or orally administered tramadol is effective in the treatment of acute postoperative pain in adults, and it can also block potassium channels. Also, the serotonin (5-hydroxytryptamine, 5-HT) subtype 3 (5-HT3) receptors situated on peripheral nerve endings and in the dorsal laminae of the spinal cord are possibly peripheral sites of analgesic action for tramadol [11]. There are many studies evaluating the effects of tramadol as an adjuvant to local anesthetics [12,13,14]. We acted with the idea that comparing the effectiveness of such interesting and effective drugs as adjuvants in regional anesthesia would be valuable and contribute to the literature.

In ultrasound-guided TAP block applications, adjuvants can be added to the local anesthetic agent in order to increase the analgesic efficiency and reduce the potential side effects and toxic effects of the drug used. This prospective, randomized study aimed to evaluate the efficacy of perineural tramadol and dexmedetomidine added to the local anesthetic solution in prolonging postoperative analgesia, maintaining stable hemodynamic parameters, and reducing intraoperative opioid and muscle relaxant consumption, postoperative side effects, and postoperative mobilization time in patients undergoing laparoscopic cholecystectomy cases.

## 2. Methods

### 2.1. Study Design

The study was initiated after Van Yuzuncu Yil University Faculty of Medicine Ethics Committee’s approval (15.12.2020/06), registered with ClinicalTrials.gov (NCT05905757). Sixty-seven patients who applied to the anesthesiology and reanimation clinic for elective laparoscopic cholecystectomy were included. The patients were informed in detail about the study and possible complications before the operation, and their verbal and written consents were obtained. All study patients were described on a 10 cm visual analog scale (VAS) preoperatively with VAS 0–2: no pain, 3–4: mild pain, 5–6: moderate pain, 7–8: severe pain, and 9–10: unbearable pain. The inclusion criteria included patients aged 20–60 years, undergoing elective laparoscopic cholecystectomy surgery, having ASA I–II physical status, and giving consent. Patients outside the 20–60 age range; with ASA III and above physical status; with severe heart, lung, or liver disease; with kidney failure; with bleeding diathesis; who developed complications during the procedure; with fever and active infection; allergic to the drugs to be used in the study; who refused to participate; who had hypothermia or acid-base disorder; and who were taking antibiotics, anticonvulsants, antiarrhythmics, or cholinesterase inhibitors; as well as pregnant women; bleeding cases; emergency cases; and those with a BMI of 30 and above were excluded from the study (Figure 1).

### 2.2. Sample Size Calculation

The purpose of adding dexmedetomidine (Sedadomid^®^, Kocak Farma, Istanbul, Turkey) or tramadol (Ramadex^®^, Haver, Tekirdag, Turkey) as an adjuvant to the local anesthetic is to observe the effect on the analgesic duration. According to the results of previous similar studies on this subject [15,16], at least 30 patients were required to detect a significant difference between the groups in terms of 180 min analgesia duration with a power of 80% and an α error of 0.05. Considering the patients who might be excluded from the study, the study started with 67 patients.

### 2.3. Anesthesia Application

A 22-gauge intravenous line was placed, and 10 mL kg^−1^ isotonic saline infusion was started. All patients were premedicated with 0.3 mg/kg midazolam. Standard monitoring with electrocardiography, non-invasive blood pressure, peripheral oxygen saturation, and bi-spectral index monitoring (Datex-Ohmeda S/5 monitor MBIS module, Helsinki, Finland) was applied to all patients in the operating room. Anesthesia induction was performed with intravenous propofol 2–3 mg kg^−1^, fentanyl 2 μg kg^−1^, and rocuronium bromide 0.6 mg kg^−1^ by the same anesthesiologist. Endotracheal intubation was performed when the BIS score was 40–60. Maintenance of anesthesia was provided with 4–6% end-tidal desflurane in 3 L of 40% O_2_ and 60% air mixture. The minimum alveolar concentration of desflurane was targeted to reach a BIS value between 40 and 60. All patients were administered 20 mg/kg paracetamol IV as a standard analgesic before extubation.

### 2.4. Patient Randomization

The patients were randomly allocated into two groups using the single-blind closed-envelope method by a researcher who was not involved in the study.

### 2.5. Block Application

Following intubation, patients received a TAP block with an oblique subcostal approach in the supine position. The same anesthesiologist placed the ultrasound (Esaote^®^ MyLab 5, Florence, Italy) probe (12MHz, linear probe; LA4 35) obliquely on the upper abdominal wall along the subcostal margin in the midline of the abdomen. The landmarks were the rectus abdominis muscle and transversus abdominis muscle. The probe was moved until the aponeurosis of the external, internal oblique, and transversus abdominis were seen. Then, the transversus abdominis muscle was identified by moving the probe laterally. The anesthesiologist directed the peripheral block needle (100 mm 22 G Echoplex, Vygon, Ecouen, France) toward the transversus abdominis fascia and injected 20 mL solution between the rectus abdominis and transversus abdominis muscles along the subcostal line. The same procedure was repeated on the contralateral side. 

Group T (Bupivacaine (Buvicaine^®^ 0.5%, Polifarma, Tekirdag, Turkey) + Adjuvant tramadol): 0.250% bupivacaine 15 mL + adjuvant 1.5 mg/kg (100 mg maximum) tramadol 25 mL, 40 mL in total, 20 mL was applied to each side.

Group D (Bupivacaine + adjuvant dexmedetomidine): 0.250% bupivacaine 15 mL + 0.5 mcg/kg and (50 mcg maximum) dexmedetomidine 25 mL, 40 mL in total, 20 mL was applied to each side.

Demographic data (age, weight, height, body mass index (BMI)), systolic arterial pressure (SAP), diastolic arterial pressure (DAP), mean arterial pressure (MAP), HR values, and operation times of the patients were recorded. Heart rate, non-invasive blood pressure, and SpO_2_ measurements were taken immediately before induction, at 15 min intervals after intubation, and 5 min after extubation. Postoperative nausea and vomiting were noted as ‘present or absent’ only once, at the sixth postoperative hour. The patients were evaluated with a 0–10 verbal rating scale visual analog scale (VAS) at 0 h (modified Aldrete score ˃ 9 in the recovery unit), 3 h, and 6 h after the operation, and the data were recorded. In the postoperative period, VAS value ˃ 4 was evaluated in favor of analgesic need in all groups, and 50 mg Dexketoprofen IV bolus was administered in accordance with conventional treatment. There was no need for additional analgesia postoperatively at 6 h.

## 3. Statistical Data Analysis

In the descriptive statistics of the data, mean, standard deviation, median minimum and maximum, frequency, and ratio values were used. The distribution of variables was measured with the Kolmogorov–Smirnov test. Independent sample *t*-test and Mann–Whitney U test were used in the analysis of quantitative independent data. The Chi-square test was used in the analysis of qualitative independent data. The SPSS 28.0 software (SPSS Inc., Chicago, IL, USA) program was used in the analysis.

## 4. Results

T and D groups did not differ significantly in terms of demographic data and ASA scores (*p* > 0.05) (Table 1).

When the groups were compared in terms of intraoperative heart rate, it was found that there was no statistically significant (*p* > 0.05) difference between the groups until the 30th minute, and the intraoperative 30th-minute values were higher in favor of the T group (*p* < 0.05) (Figure 2).

Systolic and diastolic blood pressures did not differ significantly (*p* > 0.05) between the groups for the pre-induction, intraoperative, and post-extubation periods. While the mean arterial pressures at the pre-induction and intraoperative 15th minutes did not differ between the groups (*p* > 0.05), the intraoperative 0 min and 30 min values were found to be significantly lower, in favor of the T group (*p* ˂ 0.05). The mean arterial pressure at the fifth minute after extubation was found to be significantly higher in group T (*p* ˂ 0.05) (Table 2).

VAS scores did not differ between the groups in all evaluated time periods (0 h, 3 h, and 6 h postoperatively) (*p* > 0.05) (Figure 3).

The groups were also compared in terms of additional opioid and muscle relaxant needs (fentanyl, rocuronium), postoperative non-opioid analgesic consumption, mobilization times, and side effects (nausea, vomiting, etc.), and no statistically significant difference was observed (*p* > 0.05) (Table 3).

## 5. Discussion

Transversus abdominis plane block can be applied as part of a good perioperative hemodynamic and analgesic regimen in appropriate abdominal surgeries. This situation has brought with it efforts to increase the effectiveness and duration of the action of peripheral nerve blocks made for analgesic purposes. This study aimed to evaluate the efficacy of adjuvant tramadol and dexmedetomidine in addition to bupivacaine in the TAP block procedure, which was performed to increase postoperative analgesic efficacy, reduce the incidence of side effects, reduce the need for additional intraoperative doses (neuromuscular blocker, opioid), and shorten mobilization time. The doses to be used were determined based on different studies in the literature, and the minimum dose and maximum effect were targeted. This study, which is among the few studies evaluating the adjuvant efficacy of dexmedetomidine and tramadol, showed that dexmedetomidine provided stable intraoperative hemodynamics and showed similar efficacy to tramadol in terms of analgesic efficacy.

Postoperative analgesia management is a topic that remains popular today. Inadequate use of analgesics in the treatment of pain causes unmanageable pain, and unnecessary use of analgesics causes side effects [17]. Neethirajan et al. [18] used 1 µg/kg dexmedetomidine as an adjuvant to bupivacaine in TAP block in their randomized controlled study conducted in 2020. Heart rate and blood pressure values in the dexmedetomidine group were significantly lower than in the control group. In their study, the median value for the 30th-min heart rate was 56/min, and the mean arterial pressure was 72 mmHg. Similarly, in our study, heart rate and mean arterial pressure values at the 30th minute intraoperatively and after extubation were found to be lower in D-group patients than in T-group patients. The heart rate and blood pressure values were closer to the physiological limits in the D group because of the adjuvant dexmedetomidine for the dose we used. This suggests that it provides a more stable perioperative process in terms of hemodynamics.

In a large-scale meta-analysis by Abdallah et al. [19], in which a minimum of 0.75 µg/kg dexmedetomidine was added to local anesthetic as an adjuvant in peripheral block applications, it was reported that dexmedetomidine caused transient bradycardia despite its analgesic efficacy and time advantages. It was stated that cases with bradycardia were transient and reversible with atropine. On the contrary, in our study, there were no cases of bradycardia and hypotension as complications at a dose of 0.5 µg/kg dexmedetomidine, results that were more stable in terms of hemodynamics and closer to the physiological limits that were obtained in the dexmedetomidine group.

Basavarajaiah et al. [20] applied TAP block by adding tramadol and dexmedetomidine as adjuvants to levobupivacaine in pediatric laparoscopic orchiopexy surgeries in 2022. In their study evaluating the analgesic efficacy, they added 1 µg/kg dexmedetomidine and 1 mg/kg tramadol to levobupivacaine. They used the Face, Legs, Activity, Cry, and Consolability (FLACC) pain scale in the postoperative period and showed that the postoperative analgesia time was significantly longer in the dexmedetomidine-added group. This differs from our study in that the local anesthetic doses applied, the doses of tramadol and dexmedetomidine added as adjuvants, and the pain scale used for follow-up were different. In our study, postoperative 0 h, 3 h, and 6 h VAS scores did not differ significantly (*p* > 0.05) in T- and D-group patients. It was observed that they provided more effective analgesia with the high-dose dexmedetomidine they used in their studies. In this respect, the current study is important in terms of shedding light on new studies that will provide more effective analgesia and stable hemodynamics at different doses than those used in our study. In our study, no difference was observed between the tramadol and dexmedetomidine groups in terms of side-effect profiles, in line with the study of Basavarajaiah et al.

Elyazed et al. [21] investigated the effects of tramadol and dexmedetomidine added to ropivacaine as an adjuvant in supraclavicular brachial plexus block in a study they conducted on 105 patients in 2015. Compared with the control group, the analgesia duration of the groups with tramadol and dexmedetomidine was found to be significantly longer. While their study is a peripheral nerve block study, our study is a facial plane block study. Therefore, it would not be correct to compare the peripheral nerve block with the facial plane block. Additionally, the difference in local anesthetics used is also an important issue. The purpose of sharing this study here is to draw attention to the fact that both dexmedetomidine and tramadol are used as adjuvants in the supraclavicular block, which is a peripheral nerve block.

There are some limitations to this study. The main limitation of this study is the lack of a no-intervention control group. Secondly, postoperative pain perception is a subjective phenomenon that is burdensome to quantify. Thirdly, since there are not many studies with similar features in the literature, the doses we determined are minimal doses that provide effectiveness. Working with different doses and more participants may change the results. Finally, clinical signs or symptoms of neurotoxicity were not assessed.

As a result, we found that tramadol and dexmedetomidine did not have significant differences in postoperative analgesia, postoperative analgesic consumption, side effect profile, and postoperative mobilization time when added as an adjuvant to local anesthetic drugs in TAP block applications. However, we observed that the dexmedetomidine group provided more physiological hemodynamics in terms of heart rate and mean arterial pressure. We think that adjuvant dexmedetomidine would be appropriate when applying a TAP block for more stable hemodynamic results. In addition, we believe that new studies with the use of dexmedetomidine over 50 µg as an adjuvant to increase the duration of sensory and motor blockade in adults may yield more meaningful results on the duration of postoperative analgesia.

## Figures and Tables

**Figure 1 jcm-12-07001-f001:**
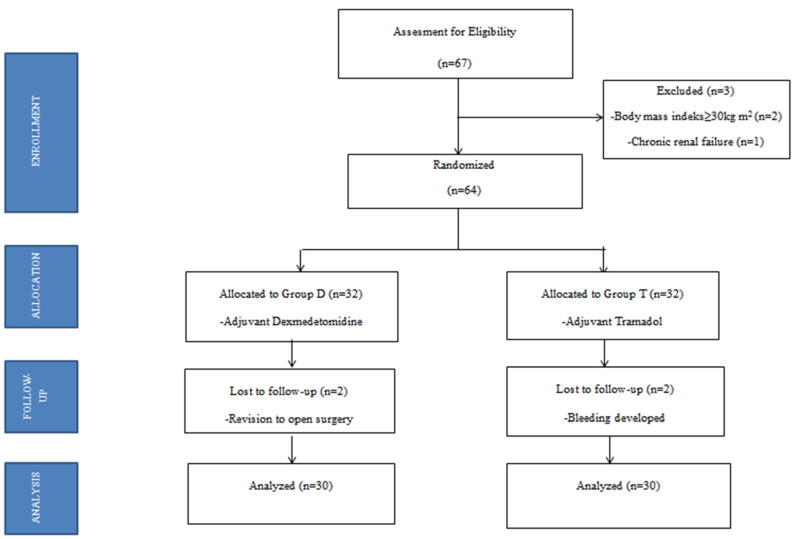
Flowchart of the study.

**Figure 2 jcm-12-07001-f002:**
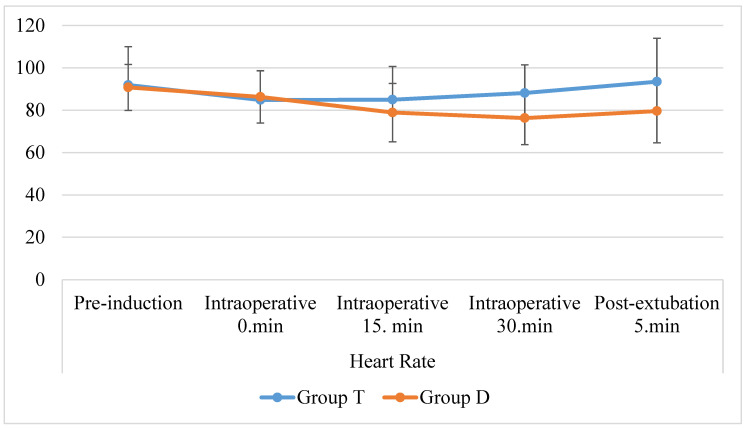
Heart rate changes.

**Figure 3 jcm-12-07001-f003:**
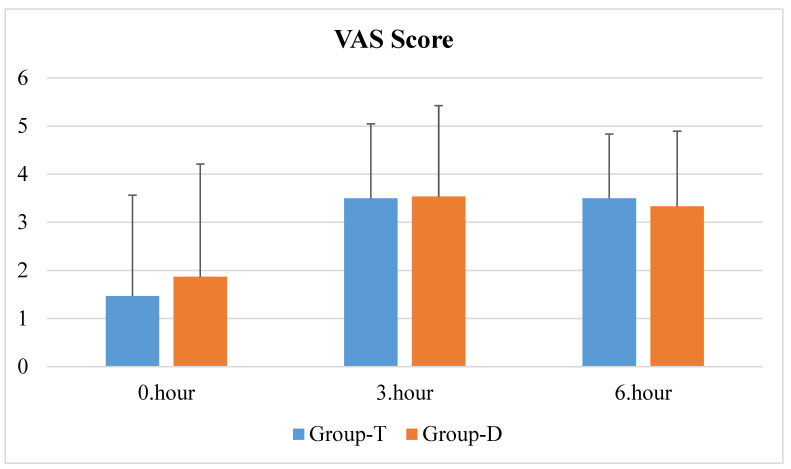
VAS scores for all groups.

**Table 1 jcm-12-07001-t001:** Demographic data.

		Group T		Group D		*p*
		Mean ± sd/n%		Mean ± sd/n%		
Age		45.9 ± 10.1		42.7 ± 10.9		0.428 ^t^
Gender	*Female*	21	70.0%	25	83.3%	0.222 ^X²^
	*Male*	9	30.0%	5	16.7%	
Height (cm)		162.7 ± 9.9		162.5 ± 7.8		0.562 ^m^
Weight (kg)		77.3 ± 15.1		74.0 ± 13.1		0.375 ^t^
BMI		29.2 ± 5.7		28.0 ± 4.8		0.371 ^m^
ASA	*I*	16	53.3%	13	43.3%	0.438 ^X²^
	*II*	14	46.7%	17	56.7%	

^t^ Independent sample *t*-test/^m^ Mann–Whitney U test/^X²^ Chi-square test.

**Table 2 jcm-12-07001-t002:** Evaluation of blood pressures.

	Group T	Group D	*p*
	Mean ± sd	Mean ± sd	
** *Systolic Blood Pressure (SBP)* **			
Pre induction	143.5 ± 19.6	143.5 ± 17.9	0.935 ^m^
Intraoperative 0 min	116.0 ± 21.2	128.1 ± 19.5	***0.011*** ^m^
Intraoperative 15 min	122.9 ± 30.3	119.3 ± 21.4	0.605 ^m^
Intraoperative 30 min	126.2 ± 21.8	112.7 ± 19.7	0.053 ^m^
Post extubation 5 min	136.1 ± 18.6	128.3 ± 17.6	0.156 ^m^
** *Diastolic Blood Pressure (DBP)* **			
Pre induction	78.9 ± 11.6	82.0 ± 13.0	0.339 ^t^
Intraoperative 0 min	70.7 ± 14.6	77.5 ± 12.8	0.058 ^t^
Intraoperative 15 min	78.9 ± 24.0	76.9 ± 17.2	0.703 ^t^
Intraoperative 30 min	78.6 ± 13.5	71.0 ± 14.5	***0.042*** ^t^
Post extubation 5 min	83.2 ± 12.6	78.5 ± 13.8	0.173 ^t^
** *Mean Arterial Pressure (MAP)* **			
Pre induction	104.8 ± 13.0	104.7 ± 14.1	0.988 ^m^
Intraoperative 0 min	89.1 ± 15.1	96.6 ± 14.2	***0.017*** ^m^
Intraoperative 15 min	96.7 ± 23.7	93.4 ± 17.4	0.451 ^m^
Intraoperative 30 min	96.4 ± 14.1	87.5 ± 15.6	***0.043*** ^m^
Post extubation 5 min	103.9 ± 14.5	96.3 ± 15.1	***0.037*** ^m^

^t^ Independent samples *t* test/^m^ Mann–Whitney U test. (Bold numbers are significant values).

**Table 3 jcm-12-07001-t003:** Additional dose requirement, analgesic consumption, mobilization times, and side effects.

		Group T		Group D		*p*
		Mean ± sd/n-%		Mean ± sd/n-%		
** *Additional dose requirement* **	(-)	22	73.3%	27	90.0%	0.095 ^X²^
	(+)	8	26.7%	3	10.0%	
** *Rocuronium (mg)* **		10.0 ± 0.0		10.0 ± 0.0		1.000 ^m^
** *Fentanyl (µcg)* **		56.3 ± 17.7		50.0 ± 0.0		0.540 ^m^
** *Postoperative consumption of nonopioid analgesics* **						
1 h	(-)	24	80.0%	26	86.7%	0.488 ^X²^
	(+)	6	20.0%	4	13.3%	
2 h	(-)	30	100.0%	30	100.0%	1.000 ^X²^
	(+)	0	0.0%	0	0.0%	
3 h	(-)	20	66.7%	20	66.7%	1.000 ^X²^
	(+)	10	33.3%	10	33.3%	
4 h	(-)	30	100.0%	30	100.0%	1.000 ^X²^
	(+)	0	0.0%	0	0.0%	
5 h	(-)	30	100.0%	30	100.0%	1.000 ^X²^
	(+)	0	0.0%	0	0.0%	
6 h	(-)	18	60.0%	20	66.7%	0.592 ^X²^
	(+)	12	40.0%	10	33.3%	
** *Postoperative mobilization time* **		4.0 ± 1.8		4.0 ± 1.4		0.862 ^m^
** *Nausea* ** ** *Vomiting* **	(-)	23	76.7%	19	63.3%	0.260 ^X²^
	(+)	7	23.3%	11	36.7%	

^m^ Mann–Whitney U test/^X²^ Chi-square test.

## Data Availability

If required, our data can be submitted.

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
