# Peer review of "Intraoperative and Postoperative Effects of Dexmedetomidine and Tramadol Added as an Adjuvant to Bupivacaine in Transversus Abdominis Plane Block"

_jcm, 2023, doi:10.3390/jcm12227001_

Round 1
Reviewer 1 Report
Comments and Suggestions for Authors
Dear authors,
Thank you for the opportunity to read and review the manuscript.
The topic is very interesting.
Please improve the manuscript based on the following comments:
General comments
Locoregional anesthesia has spread during the last few years as it can be useful to reduce time of hospitalization, adverse events related to general anesthesia and reduce pain; moreover, it can be useful to reach an, almost, opioid-free anesthesia. It can be used as substitute for general anesthesia or as adjuvant to general anesthesia to obtain better pain control and less use of intra and postoperative analgesics (most of all opioids), thus resulting in less nausea and vomiting risk, faster mobilization and recovery and more rapid hospital discharge [Mulier JP. Perioperative opioids aggravate obstructive breathing in sleep apnea syndrome: mechanisms and alternative anaesthesia strategies. Curr Opin Anaesthesiol. 2016;29(1):129–33; Ibrahim M, Elnabtity AM, Hegab A, Alnujaidi OA, El Sanea O. Combined opioid free and loco-regional anaesthesia enhances the quality of recovery in sleeve gastrectomy done under ERAS protocol: a randomized controlled trial. BMC Anesthesiol. 2022 Jan 21;22(1):29. doi: 10.1186/s12871-021-01561-w. PMID: 35062872; PMCID: PMC8781357)].
Specific comments
Abstract
English must be improved. Methods section should be improved, as I will discuss below.
Introduction
English must be improved.
Methods section
English must be improved.
Inclusion and exclusion criteria should be better explained. How did you choose the age of inclusion 20-60?
The characteristics of the two groups should be better explained in line 67-71, please improve.
Line 71: does MAK stands for MAC minimum alveolar concentration?
Line 77: words are missing, please improve.
Line 77-82: subcostal TAP block is not performed placing the probe on the line between spina iliaca anterior superior and umbilicus but with the probe parallel and caudad to the costal margin and it ideally anesthetizes intercostal nerves T6-T9 between rectus abdominis sheath and the transversus abdominis muscle and not ileoinguinal and iliohypogastric nerves. Moreover (line 80), subcostal TAP block (both upper and lower) and lateral TAP block (that ideally reach intercostal nerves T10-T11 and subcostal nerve T12) are fascial block (AL spread between rectus abdominis sheath and the transversus abdominis muscle and nerves usually could not be seen). Please improve.
Line 83-91: measurements are not clear in terms of data recording and, most of all, timing. When did you collected data? In results section you also reported heart rate and pressure values 15th minutes and 30th minutes (line 114, line 120, table 1, figure 1), after AG induction, data recordings and timings are not clear, please specify in methods section.
Results section
As aforementioned time periods should be better reported in methods section as they are not clear. Please improve.
Line 114 and Figure 1, mean (and st dev) HR values could be reported to better highlight the differences.
Table 3 is interesting.
Discussion section
English must be improved.
Line 186: I think that a supraclavicular plexus block could not be compared, in terms of drugs doses, to TAP block as the latter is a fascial block. The study of Elayazed et al is interesting, however conclusion in line 186 is not properly consistent with your study design (fascial block versus plexus block).
Limitation section is missing.
Comments on the Quality of English Language
Dear authors,
english must be improved in all the sections of the manuscripts to make it more clear and improve its quality.
Author Response
We would like to thank to reviewer for his/her kind support to our manuscript. We did all necessary corrections and sent the last version of manuscript
Reviewer 1
Q1- General comments
Locoregional anesthesia has spread during the last few years as it can be useful to reduce time of hospitalization, adverse events related to general anesthesia and reduce pain; moreover, it can be useful to reach an, almost, opioid-free anesthesia. It can be used as substitute for general anesthesia or as adjuvant to general anesthesia to obtain better pain control and less use of intra and postoperative analgesics (most of all opioids), thus resulting in less nausea and vomiting risk, faster mobilization and recovery and more rapid hospital discharge [Mulier JP. Perioperative opioids aggravate obstructive breathing in sleep apnea syndrome: mechanisms and alternative anaesthesia strategies. Curr Opin Anaesthesiol. 2016;29(1):129–33; Ibrahim M, Elnabtity AM, Hegab A, Alnujaidi OA, El Sanea O. Combined opioid free and loco-regional anaesthesia enhances the quality of recovery in sleeve gastrectomy done under ERAS protocol: a randomized controlled trial. BMC Anesthesiol. 2022 Jan 21;22(1):29. doi: 10.1186/s12871-021-01561-w. PMID: 35062872; PMCID: PMC8781357)].
A1-General comments suggested by the Reviewer were placed in the introduction part of the article.
Q2- Specific comments
Q2-a: Abstract
English must be improved. Methods section should be improved, as I will discuss below.
A2-a: It was edited in English and its certificate was added. Methods section was improved.
Q2-b: Introduction
English must be improved.
A2-b: It was edited in English and its certificate was added
Q3- Methods section
English must be improved.
A3- It was edited in English and its certificate was added
Q3-a: Inclusion and exclusion criteria should be better explained. How did you choose the age of inclusion 20-60?
A3-a: Inclusion and exclusion criteria were explained in methods section.
In our country, it is very difficult to obtain ethics committee approval and conduct studies for interventional procedures in children under the age of 18. Since the age range in laparoscopic cholecystectomy surgeries performed in our hospital is generally 20-60, and since most studies were conducted in this age group in our literature review, the age range in our study was determined as 20-60.
Q3-b: The characteristics of the two groups should be better explained in line 67-71, please improve.
A3-b: The characteristics of groups were revised
Q3-c: Line 71: does MAK stands for MAC minimum alveolar concentration?
A3-c: Yes, it means MAC. But I have written in Turkish unfortunately. I corrected it.
Q3-d-e: Line 77: words are missing, please improve.
Line 77-82: subcostal TAP block
A3-d-e: Missing words are written and Subcostal TAP block was explained with detail.
Q4-: Line 83-91: measurements are not clear in terms of data recording and, most of all, timing.
A4-: Times for all measurements were written clearly in the text.
Q5-: As a forementioned time periods should be better reported in methods section as they are not clear. Please improve.
A5-: All times were corrected but if there is a problem again, i can check as reviewer wanted.
Q6-: Line 114 and Figure 1, mean (and st dev) HR values could be reported to better highlight the differences.
A6-: I got help from a professional statistician for all tables and figures. I shared these tables and figures without making any changes. St dev parameter was added to figures
Q7-: Table 3 is interesting.
A7-:Thank you so much. Yes this table is like a summary of study.
Q8-: Discussion section
English must be improved.
A8-: It was edited in English and its certificate was added
Q9-: Line 186: I think that a supraclavicular plexus block could not be compared, in terms of drugs doses, to TAP block as the latter is a fascial block. The study of Elayazed et al is interesting, however conclusion in line 186 is not properly consistent with your study design (fascial block versus plexus block).
A9-: I would like to thank the reviewer for this nice and justified criticism. I made the necessary corrections and shared it in the relevant area.
Q10-: Limitation section is missing.
A10-:Limitation section is added at the end of the manuscript.
Reviewer 2 Report
Comments and Suggestions for Authors
Thank ou for permiting me to review this manuscript
Please provide a flowchart
Please detail he power analysis
please explain why do you compare tramadol which is an analgesic , with dxm which is mostly a sedative
please define MAK MAC,?
minimal alveolar concentration ?
Line 86 what is parol IV ?
edfine company for spss software
please format adequately the PDF line 105- 112
figure 1 and figure 2 please provide standard deviation
Line 143 TAP is not a peripheral nerve block
Comments on the Quality of English Language
the english is not fluid
Author Response
We would like to thank to reviewer 2 for his/her importatn supports to our manuscript. All necessary corrections were done and the last version of manuscript were uploaded.
Reviewer 2
Q1-: Please provide a flowchart
A1-:Flowchart was provided and presented as Figure 1.
Q2-: Please detail he power analysis
A2-: Power analysis was explained with detail in sample size calculation section.
Q3-: please explain why do you compare tramadol which is an analgesic , with dxm which is mostly a sedative
A3-: Dexmedetomidine is a highly selective α2-adrenergic agonist characterized by potent analgesic, sedative, antihypertensive, and anesthetic sparing effects. Besides, its central mediated analgesia, the mechanism by which dexmedetomidine enhance the quality of regional anesthesia when used as adjuvant to LA can be explained by two peripheral mechanisms. 1st is the vasoconstrictor effect around the site of injection which lead to delay of the absorption of the LA and prolong the duration of the LA effect. 2nd mechanism is the direct action of dexmedetomidine on the activity of peripheral nerves.
Tramadol can be used as an adjunct for regional anaesthesia for many reasons. Tramadol’s posses monoaminergic actions that include peripheral α2 agonism, suggesting a similar role in nerve blocks as clonidine. Also, the serotonin (5-hydroxytryptamine, 5-HT) subtype 3 (5-HT3) receptors situated on peripheral nerve endings and in the dorsal laminae of the spinal cord are possibly peripheral sites of analgesic action for tramadol.
Elyazed MMA, Shaimaa FM. Dexmedetomidine Versus Tramadol as Adjuvants to Ultrasound Guided Supraclavicular Brachial Plexus Block in Patients Undergoing Hand and Forearm Surgery. Department of Anesthesia and Surgical ICU, Faculty of Medicine, Tanta University.2015;13.2.
Q4-: please define MAK MAC,?
minimal alveolar concentration ?
A4-: Yes, MAK means MAC. We did the necessarry correction.
Q5-: Line 86 what is parol IV ?
A5-: It is paracetamol. We corrected in the manuscript.
Q6-: Define company for spss software
A6-: SPSS is a package program used to process and analyze data obtained from many different sources. Nowadays, SPSS program; government agencies, survey companies, educational researchers, health researchers, and data miners use it.
SPSS program was released in 1968 by SPSS Inc. as the first statistical software in the world. That is why the SPSS program makes very important contributions to the development of statistical science in the world. SPSS program was purchased by IBM in 2009. It currently continues its activities under the name IBM SPSS Statistics.
SPSS program is a Windows-based program. Nowadays, versions are also available for operating systems such as Windows and Linux. SPSS is not a programming language. SPSS stands for Statistics Package for Social Sciences. (Statistical Science for Social Sciences)
As its name suggests, although statistical analysis in social sciences was first developed to enable the processing of these analyses, making predictions for the analysis and interpreting them, today it is used more by researchers working with big data obtained from different sources at a certain time. SPSS is a program that provides analysis of statistical data. Can only be used for statistical data.
There are four software utilities that make SPSS analysis easier. These;
IBM SPSS Statistics (Spss statistical models); Used for fundamental analysis. It is used to understand the data in bivariate analyses, to describe the tables of the data, and to make accurate predictions about the analyses.
IBM SPSS Modeler (Model program); It is used to make predictions based on advanced statistical methods and to show the accuracy of predictions. It is graphical.
Visualization Designer; It is used to visually display the data used in the analysis.
Test Analytics; It is used to enter written data such as surveys into the program, to perform analyzes for the data, and to interpret and understand the analyses.
Q7-: please format adequately the PDF line 105- 112
A7-: Necessary corrections were made in the relevant area
Q8-: figure 1 and figure 2 please provide standard deviation
A8-:Standard deviation values were added figures
Q9-: Line 143 TAP is not a peripheral nerve block
A9-:The suggestion was corrected and written at the begining of discussion part.
Round 2
Reviewer 1 Report
Comments and Suggestions for Authors
Dear authors,
Thank you for the opportunity to read and review the revised version of the manuscript.
The quality of the manuscript has improved, congratulation to the authors.
Author Response
Dear Reviewer,
Thank you very much for your valuable contributions and your time. We hope that, with your help, we can produce an article that will contribute to the literature.
Reviewer 2 Report
Comments and Suggestions for Authors
The authors have mostly responded to my queries however I expect that the responses should appeear in the text
for example I know perfectly well what spss is , and there was no need to explain it to me i n a chapter , i meant that details of spss should be explained in the text , , which country , which town etc , athors should check in other manuscript
the shorthcoming of the study should appear before the conclusion not in the end
Author Response
Dear Reviewer, thank you very much for your valuable contributions. The article will gain more value thanks to you. I made all the corrections you requested. However, if there is a place that is overlooked, we would be honored by your suggestions. I tried to mark the corrected areas in yellow.
